# Characterization of an International High-Risk *Escherichia coli* ST410 Clone Coproducing NDM-5 and OXA-181 in a Food Market in China

Wan-Yun He,[a,b] Lu-Chao Lv,[a,b] Wen-Xian Pu,[a,b] Guo-Long Gao,[a,b] Zi-Lin Zhuang,[a,b] Yao-Yao Lu,[a,b] Chao Zhuo,[c] Jian-Hua Liu[a,b]

[a]Guangdong Laboratory for Lingnan Modern Agriculture, Guangzhou, China

[b]Guangdong Provincial Key Laboratory of Veterinary Pharmaceutics Development and Safety Evaluation, Key Laboratory of Zoonosis of Ministry of Agricultural and Rural Affairs, National Risk Assessment Laboratory for Antimicrobial Resistant of Microorganisms in Animals, College of Veterinary Medicine, South China Agricultural University, Guangzhou, China

[c]State Key Laboratory of Respiratory Disease, First Affiliated Hospital of Guangzhou Medical University, Guangzhou, China

Wan-Yun He and Lu-Chao Lv contributed equally to this work. Author order was determined by contribution.

**ABSTRACT** During a 2020 routine epidemiological investigation of carbapenem-resistant *Enterobacterales* at a local food market in Guangzhou, China, two *Escherichia coli* ST410 isolates coproducing NDM-5 and OXA-181 were obtained from environmental samples. Antimicrobial susceptibility testing, whole-genome sequencing, and conjugation assays were applied to identify their resistance phenotypes, phylogenetic relatedness, and genetic characteristics. Phylogenetic analysis showed that the two isolates were clonally related with only one core-genome single-nucleotide polymorphism (SNP) difference and clustered into a branch with 87 *E. coli* ST410 isolates deposited in GenBank. These 89 ST410 isolates were closely related ($\leq$51 SNPs), and most were from humans in Southeast Asian countries ($n = 47$). A Vietnamese clinical isolate collected in 2017 showed the strongest epidemiological link (seven SNPs) to the two ST410 isolates detected in this study. Complete-genome analysis revealed that the carbapenem resistance determinants $bla_{NDM-5}$ and $bla_{OXA-181}$ were located on an IncF1:A1:B49-IncQ1 plasmid and IncX3 plasmid, respectively. Conjugation experiments confirmed that the IncX3 plasmid was self-transmissible while the IncF1:A1:B49-IncQ1 plasmid was nonconjugative. BLASTn analysis indicated that the two plasmids showed high similarity to other $bla_{NDM-5}$-bearing IncF1:A1:B49-IncQ1 and $bla_{OXA-181}$-bearing IncX3 plasmids from other countries. Altogether, the high similarity of the core genomes and plasmids between the ST410 isolates found in this study and those human source isolates from foreign countries suggested the clonal spread of *E. coli* ST410 strains and horizontal transmission of $bla_{OXA-181}$-bearing IncX3 plasmids across Southeast Asian countries. Stringent sanitary management of food markets is important to prevent the dissemination of high-risk clones to the public.

**IMPORTANCE** This is the first report of an *Escherichia coli* ST410 clone that coproduces NDM-5 and OXA-181 in China. The high similarity of the core genomes and plasmids between the ST410 isolates characterized in this study and human source isolates from foreign countries strongly suggests that this ST410 lineage is an international high-risk clone, highlighting the need for continuous global surveillance of ST410 clones.

**KEYWORDS** food, environment, carbapenemase, plasmid, clonal spread

International high-risk *Escherichia coli* lineages, such as ST131, ST10, ST69, ST405, and ST38, have attracted global attention; most are extraintestinal pathogens responsible for a majority of human extraintestinal infections (1). In recent years, *E. coli* clone ST410, which is associated with resistance to clinically important antimicrobials such as fluoroquinolones, third-generation cephalosporins, and carbapenems, has been identified as a

Address correspondence to Jian-Hua Liu, jhliu21@163.com.

The authors declare no conflict of interest.

**TABLE 1** *E. coli* ST410 isolates recovered from the environment of a local food market in Guangzhou, China, in August 2020

| Strain | Source | Resistance profile[a] | Resistance genotype | Plasmid type |
|---|---|---|---|---|
| GYX208DH4E-2 | Table surface | IPM, AMP, FOX, CAZ, CQ, CTX, STR, GEN, TET, DOX, SXT, CIP | *aac(3)-IId, aac(6')-Ib-cr, aadA2, aadA5, aph(3")-Ib, aph(6)-Id,* $bla_{CMY-2}$, $bla_{NDM-5}$, $bla_{OXA-1}$, $bla_{OXA-72}$, $bla_{OXA-181}$, $bla_{TEM-1B}$, *dfrA12, dfrA17, mdf*(A)*, mph*(A)*, sul1, sul2, tet*(B) | IncF1:A1:B49-IncQ1,[b] IncX3,[c] Col(BS512), ColKP3 |
| GYX208DH6-1 | Sewage near table | IPM, AMP, FOX, CAZ, CQ, CTX, STR, GEN, APR, TET, DOX, FFC, SXT, CIP | *aac(3)-IId, aac(6')-Ib-cr, aadA2, aadA5, aph(3")-Ib, aph(6)-Id,* $bla_{CMY-2}$, $bla_{NDM-5}$, $bla_{OXA-1}$, $bla_{OXA-181}$, $bla_{TEM-1B}$, *dfrA12, dfrA17, mdf*(A)*, mph*(A)*, sul1, sul2, tet*(B) | IncF1:A1:B49-IncQ1,[b] IncX3,[c] Col(BS512), ColKP3 |
| GYX208DH5-2 | Sewage near table | IPM, AMP, FOX, CAZ, CQ, CTX, APR, TET, DOX, FFC, SXT, CIP | *aac(3)-IV, aac(6')-Ib-cr, aadA5, aph(4)-Ia,* $bla_{NDM-5}$, $bla_{OXA-1}$, *floR, dfrA17, sul1, sul2, tet*(A) | IncFII:18, IncFIB:1, IncFIC:4, IncX3 |

[a]IPM, imipenem; AMP, ampicillin; FOX, cefoxidine; CAZ, ceftazidime; CQ, cefquinome; CTX, cefotaxime; STR, streptomycin; GEN, gentamicin; APR, apramycin; TET, tetracycline; DOX, doxycycline; FFC, florfenicol; SXT, trimethoprim-sulfamethoxazole; CIP, ciprofloxacin.
[b]Location of $bla_{NDM-5}$.
[c]Location of $bla_{OXA-181}$.

successful international high-risk clone, second only to the ST131 clone, and has attracted extensive attention worldwide. Previously, the ST410 clone mainly circulated in Europe and North America, but in recent years it has been a representative and regional epidemic clone in Southeast Asian countries (2). ST410 clones have dispersed into various ecological niches worldwide, including humans, food-producing animals, companion animals, wild animals, food, and the environment (3–6). More alarming is that the ST410 sublineage B4/*H24*RxC, often carrying carbapenem resistance determinants ($bla_{OXA-181}$ and/or $bla_{NDM-5}$), is rapidly emerging (6, 7), and it is reported to be responsible for the global distribution of $bla_{NDM-5}$ and $bla_{OXA-181}$ genes (8). Of greater concern is the recent emergence of ST410 clones coproducing NDM-5 and OXA-181 in human clinical settings of several countries, such as Denmark, South Korea, Egypt, and Myanmar (9–11). Here, we characterize two ST410 strains coproducing NDM-5 and OXA-181 from the food market environment in Guangzhou, China, with a strong epidemiological link to human clinical isolates from Southeast Asian countries.

## RESULTS

*E. coli* **ST410 coproducing NDM-5 and OXA-181 recovered from a local food market environment.** In August 2020, during our routine epidemiological investigation of carbapenem-resistant *Enterobacterales* (CRE) in a local food market in Guangzhou, China, we collected four environmental swab samples from a pork stall (two from the table surface and two from sewage near the table). Four meropenem-nonsusceptible isolates (GYX208DH3-2, GYX208DH4E-2, GYX208DH5-2, and GYX208DH6-1) were recovered from two table surface samples and two sewage samples. PCR amplification showed them to be positive for the carbapenem resistance gene $bla_{NDM}$, and species identification revealed that they were *E. coli*. The antimicrobial susceptibility testing results showed that they were resistant to all $\beta$-lactams (including imipenem) but susceptible to tigecycline, colistin, fosfomycin, and amikacin (Table 1).

The four *E. coli* isolates were then subjected to short-read whole-genome sequencing, and long-read sequencing was performed on strain GYX208DH4E-2 to acquire complete genomes. Sequence analysis revealed that three isolates (GYX208DH4E-2, GYX208DH5-2, and GYX208DH6-1) belonged to the high-risk ST410 lineage, harboring multiple antimicrobial resistance genes (ARGs) and plasmid replicon types (Table 1), and strain GYX208DH3-2 belonged to high-risk ST10 lineage. GYX208DH4E-2 carried three carbapenemase genes ($bla_{NDM-5}$, $bla_{OXA-181}$, and $bla_{OXA-72}$), GYX208DH5-2 carried one ($bla_{NDM-5}$), and GYX208DH6-1 carried two ($bla_{NDM-5}$ and $bla_{OXA-181}$). Notably, GYX208DH4E-2 and GYX208DH6-1 carried identical plasmid replicon types [IncFIA, IncFIB, IncFII, IncQ1, IncX3, Col(BS512), and ColKP3] and identical resistance determinants [$bla_{NDM-5}$, $bla_{OXA-181}$, *aac(6')-Ib-cr, aadA2, dfrA17, sul1*, etc.]. However, only GYX208DH4E-2 carried $bla_{OXA-72}$.

**Two *E. coli* ST410 strains are clonally related to human source isolates from Southeast Asian countries.** To investigate the clonal relationship and trace the origin of the three carbapenemase-producing ST410 isolates, we performed phylogenetic analysis

against 1,184 ST410 isolates extracted from the GenBank database (accessed 18 May 2022) and core-genome single-nucleotide polymorphism (SNP) calculations. Isolates with ≤10 SNPs were regarded as having clonal relatedness (12). We determined that GYX208DH4E-2 differed from GYX208DH6-1 by only one SNP in the core genome, and both shared highly similar drug resistance phenotypes and genotypes as well as identical plasmid types. These findings suggest that they are clonally related, implying the clonal spread of the ST410 clone coproducing NDM-5 and OXA-181 in the food market environment. However, both were phylogenetically distinct from GYX208DH5-2 by 739 SNPs, and GYX208DH5-2 was also highly distant (168 to 16,629 SNPs) from other ST410 isolates in GenBank (see Data Sets S1 to S3 in the supplemental material).

Further, compared with 1,184 GenBank isolates, strains GYX208DH4E-2 and GYX208DH6-1 were found to be clustered into a branch with 87 GenBank isolates collected in 2016 to 2021, which differed from the reference strain GYX208DH4E-2 by 7 to 51 SNPs (Fig. S1 and Data Set S4). Most of these GenBank isolates were from Asian countries, including Thailand ($n = 44$), China ($n = 21$), Myanmar ($n = 5$), South Korea ($n = 3$), Vietnam ($n = 2$), and Cambodia ($n = 1$); a few were from other countries, including Australia ($n = 5$), United States ($n = 4$), Nigeria ($n = 1$), and Switzerland ($n = 1$). In addition, almost all were from humans, except for three of the Myanmar isolates that were from food. Sequence analysis revealed that 96.6% (84/87) of these GenBank isolates concurrently carried a $bla_{NDM-5}$ gene and three IncF-type plasmid replicons (IncFII:1, IncFIA:1, and IncFIB:49), and 74.7% (65/87) harbored an IncQ1 replicon. However, 19 of the 21 isolates from China were negative for IncQ1 replicon (Data Set S4). Among these GenBank isolates, we observed two possible outbreaks of the ST410 clone; one included 36 human isolates collected from Thailand in 2016 to 2017, and the other involved 16 isolates from clinical settings in Xuzhou, China, in 2018 to 2019, which implies an ongoing spread of ST410 clones in Asian countries in recent years. In addition, some GenBank isolates were recovered from human ascitic fluid, blood, pus, and urine (Data Set S4), suggesting that they may be pathogenic and cause various human infections.

Of note, 7 of the 87 GenBank isolates had ≤10 SNP differences with GYX208DH4E-2 and GYX208DH6-1: five from Thailand and two from Vietnam. All were human isolates, with one Vietnamese clinical isolate exhibiting the strongest epidemiological link (seven SNPs; GenBank accession number GCA_018260215.1) (Fig. 1). Apart from resistance genes $bla_{OXA-181}$ and $bla_{OXA-72}$ and plasmid replicon type IncX3, GYX208DH4-2 and GYX208DH6-1 shared the same resistance genotype with the Vietnamese clinical isolate. In addition to human source isolates, three food source (pork, water spinach, and beef) isolates from local markets in Myanmar showed high similarity with GYX208DH4-2, with 13, 16, and 22 SNPs, respectively (Fig. 1).

**The IncF1:A1:B49-IncQ1 plasmid carried $bla_{NDM-5}$ and the IncX3 plasmid carried $bla_{OXA-181}$.** The genetic elements encoding $bla_{NDM-5}$ and $bla_{OXA-181}$ in strains GYX208DH4E-2 and GYX208DH6-1 were investigated. Complete genome analysis of GYX208DH4E-2 showed that $bla_{NDM-5}$ was colocated on a 99,713-bp IncF1:A1:B49-IncQ1-type hybrid plasmid (named pHN4E2-NDM5) with 14 other ARGs, including $bla_{OXA-1}$, $bla_{TEM-1B}$, $aadA2$, $aac(6')$-$Ib$-$cr$, $sul1$, etc., and the $bla_{OXA-181}$ gene was located on a 46,420-bp IncX3-type plasmid (named pHN4E2-OXA181) (Table 1); conjugation experiments revealed that only $bla_{OXA-181}$-bearing IncX3 plasmid was conjugatable. We found that the IncF1:A1:B49-IncQ1 plasmid lacked a transferability-associated region, probably accounting for its nontransferability (Fig. 2a). In addition, the draft genome of GYX208DH6-1 had highly similar mapping results to pHN4E2-NDM5 and pHN4E2-OXA181, suggesting that $bla_{NDM-5}$ and $bla_{OXA-181}$ genes in strain GYX208DH6-1 were also located on IncF1:A1:B49-IncQ1 and IncX3 plasmids, respectively (Fig. 2a and c).

In the subsequent sequence comparison of plasmid pHN4E2-NDM5 with similar plasmids in GenBank, we found that pHN4E2-NDM5 showed a high degree of similarity (92 to 100% coverage and ≥99% identity) with 11 IncF1:A1:B49-IncQ1-type plasmids from Denmark, Egypt, Spain, Thailand, Myanmar, Cambodia, South Korea, and China. All these plasmids carried the $bla_{NDM-5}$ gene (Data Set S5) (13) and were all human-derived except for pF070-NDM5 from pork. All were confirmed to be carried by *E. coli* ST410 isolates; however, the sequence types of *E. coli* isolates carrying pC016_NDM5 and pC405_NDM5 could not

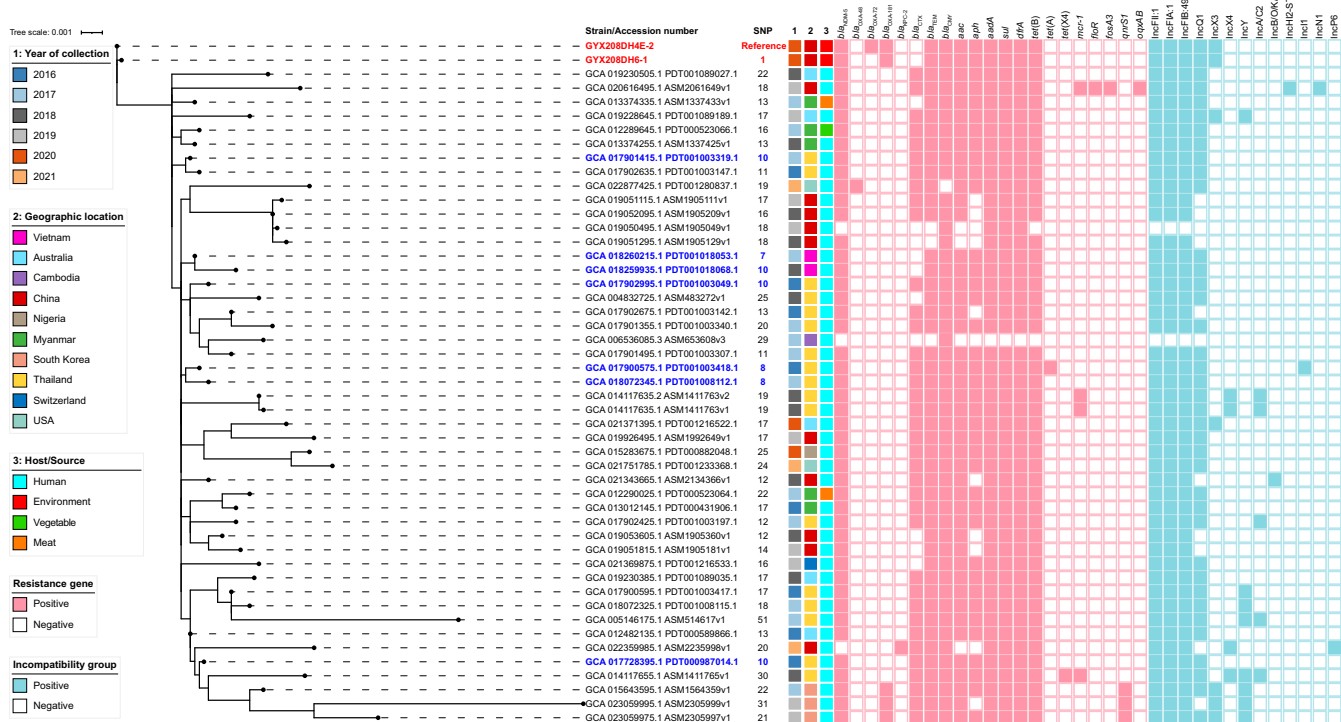

**FIG 1** Phylogenetic relatedness of *E. coli* ST410 isolates from this study and GenBank. The GenBank isolates that were clustered into a branch with *E. coli* ST410 isolates from this study (GYX208DH4-2 and GYX208DH6-1) are shown. Isolates from the same source and belonging to the same clone are represented as one isolate. *E. coli* ST410 isolates in this study are highlighted in bold red, and GenBank isolates showing strong epidemiological links (≤10 core-genome SNPs) with the reference strain GYX208DH4-2 are highlighted in bold blue. The year of collection, geographic location, and isolation source are differentiated by colors. SNP, single-nucleotide polymorphism.

be identified. Among these plasmids, pYJ1-NDM5 (GCA_013374255.1), pF070-NDM5 (GCA_013374335.1), pEC195-93k (GCA_020616495.1), and pNIPH17_0020_1 (GCA_006536085.3) were carried by the ST410 isolates belonging to the ST410 branch mentioned above (Fig. 1; Data Set S4). Importantly, except for pNIPH17_0020_1, pF070-NDM5, and pEC195-93k, the other eight human source IncF1:A1:B49-IncQ1-type plasmids shared highly identical (100% coverage and ≥99% identity) multiresistance regions with pHN4E2-NDM5 (Fig. 2b). However, compared with pAMA1167-NDM-5, the $bla_{CTX-M-15}$ gene in pHN4E2-NDM5 was truncated by an insertion sequence (IS*26*), and an 8,486-bp fragment (Tn*21*-$bla_{TEM-1}$-IS*26*-ΔIS*Ecp1*) was lost, which was probably mediated by IS*26*. Similarly, plasmid pHN4E2-OXA181 bearing $bla_{OXA-181}$ also shared a highly similar backbone and the variable region (with 100% coverage and ≥99% identity) with other $bla_{OXA-181}$-bearing IncX3 plasmids in GenBank, including seven from humans in Denmark, Spain, Myanmar, South Korea, Lebanon, and China, two from companion animals in Switzerland and Portugal, and one from the environment in Switzerland (Data Set S6). However, unlike the pHN4E2-NDM5-like GenBank plasmids, which were all from *E. coli*, those similar to plasmid pHN4E2-OXA181 were present in *Enterobacteriales* other than *E. coli* and have been isolated from various sources: humans, companion animals, and the environment (Fig. 2c). $bla_{OXA-181}$ is often carried by an IncX3 plasmid and is linked to plasmid-mediated quinolone resistance gene *qnrS1* (14); however, pHN4E2-OXA181 was lacking the *qnrS1* gene (Fig. 2d).

## DISCUSSION

CRE have been a global concern that threatens the effectiveness of last-resort antimicrobials in clinical practice. During our routine monitoring of CRE in a local food market in Guangzhou, China, we detected two ST410 *E. coli* isolates coharboring $bla_{NDM-5}$ and $bla_{OXA-181}$ genes from the environment of pork sales. Currently, only a limited number of human clinical studies have reported the $bla_{OXA-181}$ gene in *Enterobacterales* species in China, including *E. coli* (14), *Klebsiella pneumoniae* (15), *Morganella morganii* (16), and *Pseudocitrobacter faecalis* (17).

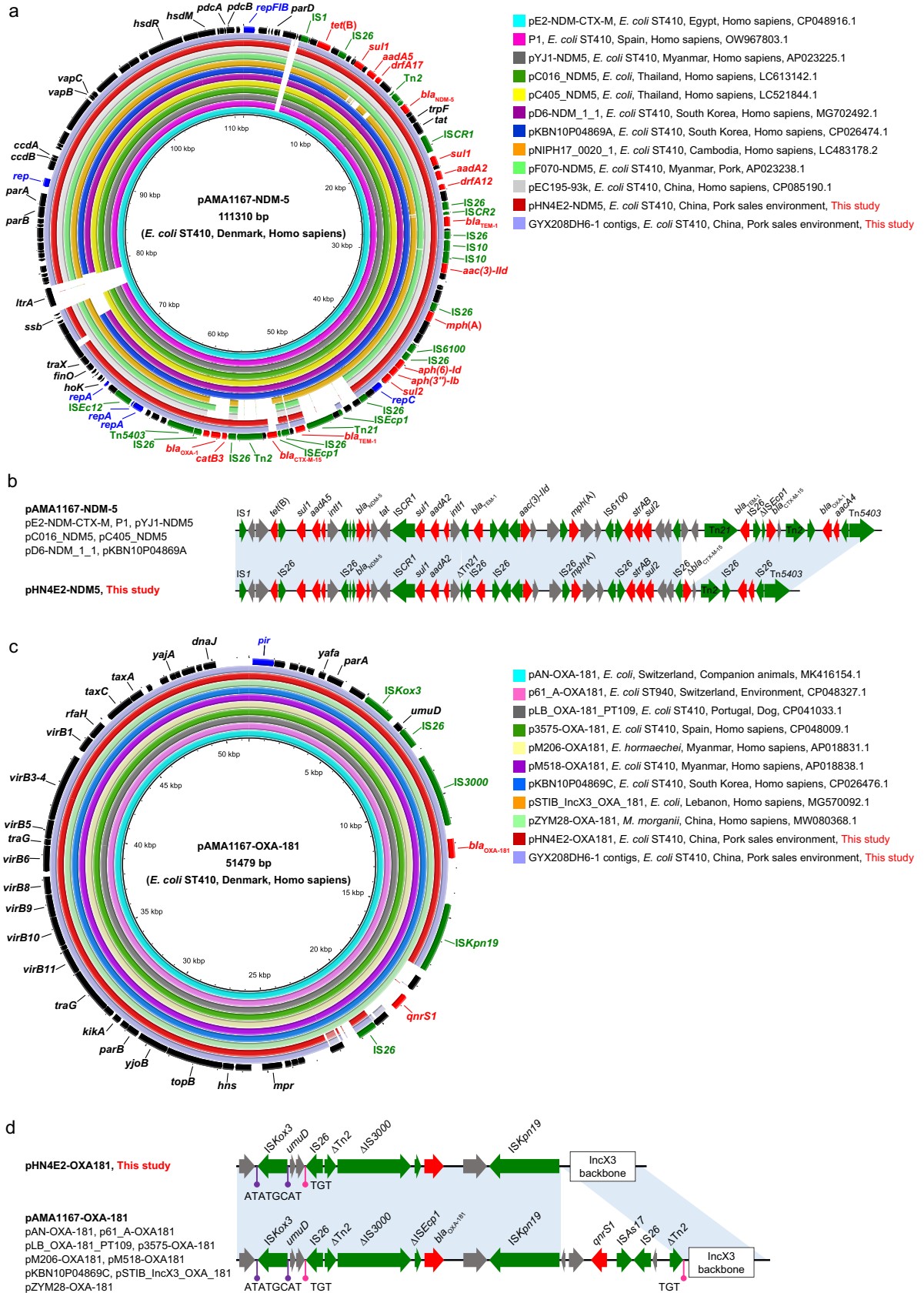

**FIG 2** Sequence comparison of IncF1:A1:B49-IncQ1-type plasmids and IncX3-type plasmids from this study and GenBank. (a) Plasmid pHN4E2-NDM5 and other IncF1:A1:B49-IncQ1 plasmids, using plasmid pAMA1167-NDM-5 (GenBank accession number CP024805.1) as a reference.

Additionally, one study reported the $bla_{OXA-181}$ gene was isolated from companion animals in the species *K. pneumoniae* (18). Here, for the first time, we discovered an environmental source $bla_{OXA-181}$ gene and reported ST410 clones coproducing NDM-5 and OXA-181 in China.

Phylogenetic analysis revealed that the two ST410 isolates we detected were clonally related (1 SNP), and both were closely related ( ≤10 SNPs) with seven GenBank isolates. All seven isolates were human-derived and from Southeast Asian countries (Vietnam and Thailand) between 2016 and 2018. Given that the two ST410 isolates in this study were more closely related to the isolates from foreign countries than the isolates from China, we therefore speculated that the ST410 clone identified in the local food market in 2020 may have recently spread across the border from human clinics (most likely in Southeast Asian countries), possibly by migration of wild birds (19), frequent trade between Guangzhou and Southeast Asia, or personnel exchanges. Of concern is that the ST410 lineage has been reported to have a competitive advantage in colonizing mammalian hosts (7), and the finding that the ST410 isolates we characterized showed strong correlations with human-sourced isolates provides further evidence for their ability to survive in the human environment. In addition, the two ST410 isolates were slightly distinct (13 to 22 SNPs) from three isolates from market food in Myanmar. These ST410 clones have the potential to infect humans via the food chain, emphasizing the importance of improving sanitation management in food markets.

Further, complete genome analysis confirmed that $bla_{NDM-5}$ and $bla_{OXA-181}$ genes in the two ST410 isolates were located on IncF1:A1:B49-IncQ1 (pHN4E2-NDM5) and IncX3 (pHN4E2-OXA181) plasmids, respectively. The plasmid pHN4E2-NDM5 was found to be highly similar to several $bla_{NDM-5}$-bearing IncF1:A1:B49-IncQ1-type plasmids from GenBank, most of which are carried by human isolates from foreign countries that belong to the aforementioned ST410 branch. Although it is unknown whether the other closely related *E. coli* ST410 strains carry the pHN4E2-NDM5-like IncF1:A1:B49-IncQ1-type plasmid, considering that most of them carried $bla_{NDM-5}$ gene and IncF-type plasmid replicons (IncFII:1, IncFIA:1, and IncFIB:49), as well as IncQ1 replicon (Fig. 1), we suspect that this *E. coli* ST410 lineage might be accompanied by the pHN4E2-NDM5-like IncF1:A1:B49-IncQ1-type plasmid. However, compared with these similar IncF1:A1:B49-IncQ1-type plasmids, e.g., pAMA1167-NDM-5, a fragment containing the $bla_{CTX-M-15}$ gene, was absent in pHN4E2-NDM5. These results suggest that the origin of pHN4E2-NDM5 is based on a pAMA1167-NDM-5-like plasmid. Consequently, the two ST410 isolates carrying pHN4E2-NDM5 in the local food market may have emerged after the human source isolates, presumably as a result of contamination by foreign isolates. Since studies have revealed the predominance of IncF1:A1:B49 plasmids encoding $bla_{NDM-5}$ and/or $bla_{CTX-M-15}$ in ST410 isolates (2, 6, 20), the role of this plasmid type in the global dissemination of ST410 clones should be investigated in the future.

Likewise, plasmid pHN4E2-OXA181 was also highly similar to some $bla_{OXA-181}$-bearing IncX3 plasmids from GenBank. Nevertheless, of the 87 phylogenetically related GenBank ST410 isolates, including those in China, most did not harbor the $bla_{OXA-181}$-carrying IncX3 plasmid (Fig. 1). To the best of our knowledge, the $bla_{OXA-181}$-carrying IncX3 plasmid is mainly prevalent in European and North American clinics (2, 6, 20), with sporadic reports in Chinese clinical settings (14–16). Therefore, the self-transmissible $bla_{OXA-181}$-carrying IncX3 plasmid pHN4E2-OXA181 may have spread from foreign countries via horizontal transfer.

In conclusion, we report here the first detection of ST410 *E. coli* carrying both $bla_{NDM-5}$ and $bla_{OXA-181}$ in the environment in China. Our findings of the high similarities of the core genomes of ST410, an IncF1:A1:B49-IncQ1 plasmid carrying $bla_{NDM-5}$, and an IncX3 plasmid carrying $bla_{OXA-181}$ between the isolates in this study and human source isolates from foreign countries strongly suggest that the ST410 isolates detected in pork sales environments may

**FIG 2** Legend (Continued)

(b) Multiresistance region of IncF1:A1:B49-IncQ1 plasmids. (c) Plasmid pHN4E2-OXA181 and other IncX3 plasmids, using plasmid pAMA1167-OXA-181 (GenBank accession number CP024806.1) as a reference. (d) Multiresistance region of IncX3 plasmids. The red, green, and blue arrows represent antimicrobial resistance genes, mobile elements, and plasmid replicons, respectively. Blue blocks represent homologous nucleotide sequences (≥99% identity). The rods represent direct repeat sequences.

originate from human isolates from foreign countries, most likely from Southeast Asia. International high-risk clones of *E. coli* ST410 producing NDM-5 and/or OXA-181 may continually spread to various geographic locations and ecological niches, eventually causing human infections. This highlights the need for concerted efforts for the continuous surveillance of ST410 clones on a global scale, and control measures such as thorough sanitary disinfection of food markets to prevent the potential spread of such high-risk clones to the public are of the utmost importance.

## MATERIALS AND METHODS

**Bacterial isolates, $bla_{NDM}$ detection, and species identification.** Samples were cultured in 5 mL Luria-Bertani (LB) broth at 37°C for 16 to 18 h, followed by bacterial isolation using MacConkey agar plates containing meropenem (1 mg/liter). PCR was performed on the isolates to screen for the carbapenem resistance gene $bla_{NDM}$, using primers previously reported (21). Species identification was performed on the $bla_{NDM}$-positive isolates using matrix-assisted laser desorption ionization–time-of-flight mass spectrometry (Shimadzu-Biotech Corp., Kyoto, Japan).

**Antimicrobial susceptibility testing.** Based on the Clinical and Laboratory Standards Institute (CLSI) guidelines (M07-A11), antimicrobial susceptibilities of isolates against 19 antimicrobials, including $\beta$-lactams (imipenem, ampicillin, cefoxitin, ceftazidime, cefquinome, and cefotaxime), aminoglycosides (streptomycin, gentamicin, apramycin, neomycin, and amikacin), tetracyclines (tetracycline, doxycycline, and tigecycline), colistin, florfenicol, fosfomycin, trimethoprim-sulfamethoxazole, and ciprofloxacin, were determined. The agar dilution method was used, except for colistin and tigecycline, got which broth microdilution was used; the quality control was *E. coli* ATCC 25922. The results were interpreted based on the criteria in CLSI (M100-S30) and EUCAST (http://www.eucast.org/clinical_breakpoints/).

**Whole-genome sequencing and analyses.** Genomic DNA was extracted from the isolates using the HiPure bacterial DNA kit (Magen Biotechnology Co. Ltd., China), according to the manufacturer's instructions. The DNA was then sequenced by short-read sequencing, using the Illumina HiSeq 2500 platform, and long-read sequencing, using Oxford Nanopore MinION (Oxford Nanopore Technologies, United Kingdom). The short reads were *de novo* assembled into contigs using SPAdes v3.8.7 (https://github.com/ablab/spades). Short and long reads were subjected to hybrid assembly using Unicycler v0.4.7 (https://github.com/rrwick/Unicycler).

The assembled contigs and genomes were submitted to the Center for Genomic Epidemiology (http://www.genomicepidemiology.org) to identify ARGs, plasmid replicons with plasmid multilocus sequencing typing, and bacterial sequence types. Core-genome SNPs were calculated using Snippy software v4.6.0 (https://github.com/tseemann/snippy). A core-genome SNP-based phylogenetic tree was constructed using Parsnp v1.5.4 (https://github.com/marbl/parsnp) and annotated using iTOL v6.5.8 (https://itol.embl.de). Sequence comparisons were performed using BRIG v0.95 (https://sourceforge.net/projects/brig/) and Easyfig v2.1 (http://mjsull.github.io/Easyfig/).

**Conjugation experiments.** The conjugative transfer abilities of carbapenemase genes $bla_{NDM-5}$ and $bla_{OXA-181}$ were assessed using LB broth mating, with ST410 isolates as the donor and sodium azide-resistant *E. coli* J53 as the recipient. Using MacConkey agar plates supplemented with 150 mg/liter sodium azide and 0.5 mg/liter meropenem, transconjugants were obtained and amplified by PCR to confirm the transfer of $bla_{NDM-5}$ and/or $bla_{OXA-181}$ by using previously described primers (14, 21).

**Data availability.** The draft genomes of strains (GYX208DH4E-2, GYX208DH5-2, and GYX208DH6-1) have been deposited in GenBank under BioProject number PRJNA869915. The complete sequences of plasmids pHN4E2-NDM5 and pHN4E2-OXA181 have been submitted to GenBank under accession numbers CP104852 and CP104853, respectively.

## SUPPLEMENTAL MATERIAL

Supplemental material is available online only.
**SUPPLEMENTAL FILE 1**, XLS file, 0.3 MB.
**SUPPLEMENTAL FILE 2**, PDF file, 1.6 MB.

## ACKNOWLEDGMENTS

We declare no conflicts of interest.

This work was supported by the National Natural Science Foundation of China (numbers 32141002 and 31625026), Laboratory of Lingnan Modern Agriculture Project (number NT2021006), and the Innovation Project of the Education Department of Guangdong Province (number 2019KQNCX008).

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
