## [Reviewer comments · Microbiology Spectrum]

Microbiology Spectrum

Characterization of an international high-risk clone *Escherichia coli* ST410 co-producing NDM-5 and OXA-181 in a food market in China

Wanyun He, Luchao Lv, WenXian Pu, Guolong Gao, Zilin Zhuang, Yaoyao Lu, Chao Zhuo, and Jian-Hua Liu

Corresponding Author(s): Jian-Hua Liu, South China Agricultural University College of Veterinary Medicine

Review Timeline:

Submission Date:	November 18, 2022
Editorial Decision:	March 14, 2023
Revision Received:	April 2, 2023
Accepted:	April 15, 2023

Editor: Zhangqi Shen

Reviewer(s): Disclosure of reviewer identity is with reference to reviewer comments included in decision letter(s). The following individuals involved in review of your submission have agreed to reveal their identity: Juan Wang (Reviewer #2)

Transaction Report:

DOI: <https://doi.org/10.1128/spectrum.04727-22>

March 14, 2023

Prof. Jian-Hua Liu
South China Agricultural University College of Veterinary Medicine
Wushan, Tianhe District
Guangzhou
China

Re: Spectrum04727-22 (**Emergence of an international high-risk clone *Escherichia coli* ST410 co-producing NDM-5 and OXA-181 in food market in China**)

Dear Prof. Jian-Hua Liu:

Link Not Available

Sincerely,

Zhangqi Shen

Journals Department
Reviewer comments:

Reviewer #1 (Comments for the Author):

The present study He et al, describes the detection of ST410 *E. coli* carrying both blaNDM-5 and blaOXA-181 in the environment in China.

Detection of anti-microbial resistant microorganisms in the environment is interesting. However, strains with high similarity of the core genomes of ST410, IncF1:A1:B49-IncQ1 plasmid carrying blaNDM-5, and IncX3 plasmid carrying blaOXA-181 between the isolates in this study and clinical strains from other countries have already been described. Even though the authors claim that the ST410 isolates detected in pork sales environments may originate from human isolates from other countries, most likely from

Southeast Asia, the study lacks evidence to suggest for that.

The study would benefit screening the individuals working in the same market (probably too late for that) to show possible spread to human beings. Is there any data on screening of individuals who worked with or stayed near the find? I.e. meat sellers or customers who have bought from there? Have the authors examine the farms where the animals come from and see if the same clone is found there? These attempts would make the study and the conclusions stronger.

It could be relevant to describe the screening process more in detail. Is it part of a general screening of food markets in China?

Reviewer #2 (Comments for the Author):

Comments for the Author :

He et al. analysed two E. coli ST410 isolates co-producing NDM-5 and OXA-181 from environmental samples in China. The genetic comparisons have been deposited and similarity have been identified with the ST410 strains mainly from human sources from other countries. These results are in my opinion interesting in order to obtain better insights in the spread of E. coli ST410 carrying antimicrobial resistance genes. Nevertheless, the manuscript needs some important revision to make it clearer and allow a complete comprehension. I also suggest that the authors have their manuscript checked by a native English speaker. Below are my remarks:

Title: Considering only two ST410 (co-producing NDM-5 and OXA-181) isolates identified in the study, I do not think the title of the manuscript is proper, especially the word "Emergence". "Genetic comparison" or "Characterization" maybe better. Also on Line 68, it's better to revise the writing "the first emergence...".

Abstract: the abstract is difficult to follow and gives not much information that can be interesting for the reader to read this paper. I suggest to re-write the abstract giving more information so that it will be more attractive to read the whole paper.

About the Vietnamese clinical isolate, no description about its plasmids?

Line 2, in the title "market" please revised as "markets";

Line 23, please revised the "a farmers' market"; and also double check the manuscript for the description about "food markets" or "a farmers' market"? Maybe "a local food market" is a better consistent description.

Line 24, "The present study aimed to characterize the two isolates." I do not think this sentence is proper here.

Line 26, "conjugation assay" please change as "conjugation assays"

Line 70, "This has provoked our interest given this clone's global spread and potential to infect humans via the food chain." Please delete this sentence.

Line 108, "in the animal food sales environment"? It is a wrong description, please correct.

Line 74-93, four meropenem-non-susceptible isolates have been sequenced. Three are belonging to ST410. How about the fourth one? It's better to give a mention of its sequence type during the description.

And also, only one isolate, GYX208DH4E-2, has been deposited for the WGS by long-read sequencing. "GYX208DH5-2 carried one (blaNDM-5), and GYX208DH6-1 carried two (blaNDM-5 and blaOXA-181)" only for short-read WGS, suggest to acquire complete genomes for these two as well.

Since lacking of the complete genomes of the two strains, it's uneasy to follow the Figure 2. For example, I noticed in the Fig. 2a, "GYX208DH6-1 contigs" were deposited for comparison with the blaNDM-5-carrying plasmids. Couldn't tell where is the GAPS happened on the contigs? The same questions for Fig. 2c. It's better to acquire the WGS for the two strains, and then make the comparison.

Staff Comments:

Preparing Revision Guidelines

Please return the manuscript within 60 days; if you cannot complete the modification within this time period, please contact me. If you do not wish to modify the manuscript and prefer to submit it to another journal, please notify me of your decision immediately so that the manuscript may be formally withdrawn from consideration by Microbiology Spectrum.

Response to Reviewers

Reviewer comments:

Reviewer #1 (Comments for the Author):

The present study He et al, describes the detection of ST410 E. coli carrying both blaNDM-5 and blaOXA-181 in the environment in China.

Detection of anti-microbial resistant microorganisms in the environment is interesting. However, strains with high similarity of the core genomes of ST410, IncF1:A1:B49-IncQ1 plasmid carrying blaNDM-5, and IncX3 plasmid carrying blaOXA-181 between the isolates in this study and clinical strains from other countries have already been described. Even though the authors claim that the ST410 isolates detected in pork sales environments may originate from human isolates from other countries, most likely from Southeast Asia, the study lacks evidence to suggest for that.

The study would benefit screening the individuals working in the same market (probably too late for that) to show possible spread to human beings. Is there any data on screening of individuals who worked with or stayed near the find? I.e. meat sellers or customers who have bought from there? Have the authors examine the farms where the animals come from and see if the same clone is found there? These attempts would make the study and the conclusions stronger.

Answer: We agree with the reviewer. However, since meat sellers and customers were unwilling to cooperate with us, there was no data on them. And since we could not trace the origin of the meat-producing animals, we did not examine the farms where the animals come from.

It could be relevant to describe the screening process more in detail. Is it part of a general screening of food markets in China?

Answer: Yes, it is part of our routine screening of food markets in Guangzhou, China. We have described this clearer in the manuscript (Lines 76-78).

Reviewer #2 (Comments for the Author):

Comments for the Author:

He et al. analysed two E. coli ST410 isolates co-producing NDM-5 and OXA-181 from environmental samples in China. The genetic comparisons have been deposited and similarity have been identified with the ST410 strains mainly from human sources from other countries. These results are in my opinion interesting in order to obtain better insights in the spread of E. coli ST410 carrying antimicrobial resistance genes. Nevertheless, the manuscript needs some important revision to make it clearer

and allow a complete comprehension. I also suggest that the authors have their manuscript checked by a native English speaker. Below are my remarks:

Answer: The manuscript has been carefully revised to make it clearer and more comprehensible, and the manuscript has been checked by a native English speaker.

Title: Considering only two ST410 (co-producing NDM-5 and OXA-181) isolates identified in the study, I do not think the title of the manuscript is proper, especially the word "Emergence". "Genetic comparison" or "Characterization" maybe better. Also on Line 68, it's better to revise the writing "the first emergence...".

Answer: We agree with the reviewer. We have revised "Emergence" as "Characterization" in the title and rewritten the sentence in Lines 69-71.

Abstract: the abstract is difficult to follow and gives not much information that can be interesting for the reader to read this paper. I suggest to re-write the abstract giving more information so that it will be more attractive to read the whole paper.

Answer: We have rewritten the abstract to make it more structural and attractive.

About the Vietnamese clinical isolate, no description about its plasmids?

Answer: Due to only the draft genome of the Vietnamese clinical isolate being available in the GenBank database, we could not obtain the complete sequences of its plasmids.

Line 2, in the title "market" please revised as "markets";

Answer: We have revised "food market" as "a food market" in the title due to the two ST410 isolates from the same food market.

Line 23, please revised the "a farmers' market"; and also double check the manuscript for the description about "food markets" or "a farmers' market"? Maybe "a local food market" is a better consistent description.

Answer: We agree with the reviewer. We have revised the "farmers' market" as "local food market" throughout the manuscript.

Line 24, "The present study aimed to characterize the two isolates." I do not think this sentence is proper here.

Answer: We agree with the reviewer. We have deleted this sentence (Line 23).

Line 26, "conjugation assay" please change as "conjugation assays"

Answer: We have revised the "conjugation assay" as "conjugation assays" (Line 24).

Line 70, "This has provoked our interest given this clone's global spread and potential to infect humans via the food chain." Please delete this sentence.

Answer: We have deleted this sentence (Line 72).

Line 108, "in the animal food sales environment"? It is a wrong description, please

correct.

Answer: We have revised this sentence (Lines 109-111).

Line 74-93, four meropenem-non-susceptible isolates have been sequenced. Three are belonging to ST410. How about the fourth one? It's better to give a mention of its sequence type during the description.

Answer: We have added the description of the fourth one in the manuscript (Lines 92-93).

And also, only one isolate, GYX208DH4E-2, has been deposited for the WGS by long-read sequencing. "GYX208DH5-2 carried one (blaNDM-5), and GYX208DH6-1 carried two (blaNDM-5 and blaOXA-181)" only for short-read WGS, suggest to acquire complete genomes for these two as well.

Answer: Since only one core-genome SNP between GYX208DH4E-2 and GYX208DH6-1 (suggesting they were clonal), performing long-read sequencing on GYX208DH4E-2 can be representative. And since GYX208DH5-2 was highly distant from GYX208DH4E-2, GYX208DH6-1, and GenBank isolates, we did not further focus on GYX208DH5-2. Therefore, we think that long-read sequencing on GYX208DH5-2 and GYX208DH6-1 is not critical for this study.

Since lacking of the complete genomes of the two strains, it's uneasy to follow the Figure 2. For example, I noticed in the Fig. 2a, "GYX208DH6-1 contigs" were deposited for comparison with the blaNDM-5-carrying plasmids. Couldn't tell where is the GAPs happened on the contigs? The same questions for Fig. 2c. It's better to acquire the WGS for the two strains, and then make the comparison.

Answer: Given that we successfully located the carbapenemase genes of GYX208DH6-1 referring to the complete genome of GYX208DH4E-2, and GYX208DH5-2 was not the focus of this study, we consider it unimportant to carry out long-read sequencing on the two strains.

April 15, 2023

Prof. Jian-Hua Liu
South China Agricultural University College of Veterinary Medicine
Wushan, Tianhe District
Guangzhou
China

Re: Spectrum04727-22R1 (**Characterization of an international high-risk clone *Escherichia coli* ST410 co-producing NDM-5 and OXA-181 in a food market in China**)

Dear Prof. Jian-Hua Liu:

Your manuscript has been accepted, and I am forwarding it to the ASM Journals Department for publication. You will be notified when your proofs are ready to be viewed.

Sincerely,

Zhangqi Shen
Editor, Microbiology Spectrum
